# A DNA assembly toolkit to unlock the CRISPR/Cas9 potential for metabolic engineering

Tigran V. Yuzbashev [1,5✉], Evgeniya Y. Yuzbasheva[2], Olga E. Melkina[3], Davina Patel [1], Dmitrii Bubnov [3], Heiko Dietz[4] & Rodrigo Ledesma-Amaro [1✉]

CRISPR/Cas9-based technologies are revolutionising the way we engineer microbial cells. One of the key advantages of CRISPR in strain design is that it enables chromosomal integration of marker-free DNA, eliminating laborious and often inefficient marker recovery procedures. Despite the benefits, assembling CRISPR/Cas9 editing systems is still not a straightforward process, which may prevent its use and applications. In this work, we have identified some of the main limitations of current Cas9 toolkits and designed improvements with the goal of making CRISPR technologies easier to access and implement. These include 1) A system to quickly switch between marker-free and marker-based integration constructs using both a Cre-expressing and standard *Escherichia coli* strains, 2) the ability to redirect multigene integration cassettes into alternative genomic loci via Golden Gate-based exchange of homology arms, 3) a rapid, simple in-vivo method to assembly guide RNA sequences via recombineering between Cas9-helper plasmids and single oligonucleotides. We combine these methodologies with well-established technologies into a comprehensive toolkit for efficient metabolic engineering using CRISPR/Cas9. As a proof of concept, we developed the *YaliCraft* toolkit for *Yarrowia lipolytica*, which is composed of a basic set of 147 plasmids and 7 modules with different purposes. We used the toolkit to generate and characterize a library of 137 promoters and to build a de novo strain synthetizing 373.8 mg/L homogentisic acid.

[1] Department of Bioengineering, Imperial College London, London SW7 2AZ, UK. [2] BioMediCan Inc., 40471 Encyclopedia Circle, Fremont 94538 CA, USA. [3] NRC 'Kurchatov Institute'—GosNIIgenetika, Kurchatov Genomic Centre, 1-st Dorozhny Pr., 1, Moscow 117545, Russia. [4] Kaesler Research Institute, Kaesler Nutrition GmbH, Fischkai 1, 27572 Bremerhaven, Germany. [5] Present address: Plant Sciences and the Bioeconomy, Rothamsted Research, West Common, Harpenden, Hertfordshire AL5 2JQ, UK. ✉email: t.yuzbashev@gmail.com; r.ledesma-amaro@imperial.ac.uk

The development of CRISPR/Cas9-based technologies has enabled quick, precise and scarless genomic modifications, which shows a great potential for microbial strain design and bioproduction. In particular, the metabolic engineering of yeasts is one of the fastest growing areas in engineering biology, enabling the sustainable production of chemicals, fuels, materials, foods and pharmaceuticals[1]. While having the large metabolic potential of eukaryotic cells, due to their single cell nature and rapid growth, yeasts are easier to engineer and cultivate at scale.

Because of that, CRISPR systems have been developed for yeasts[2–4]. One of the most important advantages of CRISPR is that, due to its high efficiency, it allows markerless genomic modifications. During traditional strain engineering, the elimination of selectable markers from the genome, known as marker recovery, is a common bottleneck[5]. Even fast-growing yeast cultures require at least five days for marker recovery[6], while integrative transformation itself takes only two or three. Therefore, in the past years several marker-free integration techniques facilitated by CRISPR have been developed[6–9]. In spite of these developments, many current metabolic engineering projects still rely on marker-based approaches, which may limit or delay success in strain optimization. In this work, we have identified 3 improvements of CRISPR/Cas9 systems which may facilitate the use of Cas9-based yeast engineering, (1) the easy swap between marker and markerless modifications (2) the quick exchange of homology arms to target different integration locations and (3) an easy method to clone gRNAs.

**Convenient return to marker-based integration**. During CRISPR-based marker-free integration, the only selective factor is the double strand break (DSB) induced by Cas9. To proliferate, cells need to repair this lesion. This may happen either by using a template (donor) that is integrated via homologous recombination (HR) or by non-homologous end-joining (NHEJ), i.e. without integration. NHEJ activity is observed in most fungal species including baker's yeast *Saccharomyces cerevisiae*[5,10]. In species where NHEJ is the predominant mechanism, a common strategy to enhance HR is to delete NHEJ genes (*KU70, KU80, POL4,* and *DNL4*)[11]. This strategy, while improving HR, does not fully prevent the undesired NHEJ activity[12]. Therefore, the isolation of transformants with modifications leading to slow growth phenotypes represents a general issue with marker-free integration approaches. This is because indel mutants produced by NHEJ can overgrow and mask rare and slow-growing integrative transformants. Although, unsuccessful engineering efforts are rarely published, there is one example related to the disruption of the *SDH5* gene in *Yarrowia lipolytica*. Despite multiple attempts, this modification was not isolated using a CRISPR-based marker-free strategy[13], whereas the successful deletion was obtained using an auxotrophic marker[14]. Similar observations are widely common in yeast engineering works where often some loci are easier to modify than others, with some remaining unsuccessful—in part due to the difficulty to select slow growers. Such mutations with deleterious effects on cell growth are frequent in metabolic engineering[15]. Genome-wide studies of *S. cerevisiae* revealed that the overexpression of 20% of the genes[16] and disruption of 15% of the non-essential genes[17] negatively affect cell growth. In addition, cell viability may be adversely affected by epistatic interactions between modifications in non-allelic genes[18]. Therefore, engineering approaches based on marker-free integration may involve occasional (but hard to predict) cases where stringent selection based on an auxotrophic marker is needed. In these cases, a new cassette would need to be reassembled, significantly delaying the progress. A desired improvement to CRISPR/Cas9-assisted integration systems would be therefore a method to easily revert to marker-based integration when the marker-free attempt fails.

**Redirection of donor DNA**. Cas9-assisted integration requires a donor template which consists of an integration cassette flanked by two homology arms (HAs). Each pair of HAs targets a unique genome locus and they can only be used once in a given strain for the overexpression of a limited number of transcription units (TUs). As synthetic biology advances, the average number of modifications per strain increases. Therefore, the number of genomic sites suitable for homologous integration must grow in parallel. Accordingly, sets of ready to use donor vectors with different HAs have been developed for most industrially relevant yeasts such as *S. cerevisiae*[19], *Ogataea polymorpha*[20], *Kluyveromyces marxianus*[21], *Komagataella phaffii*[22], and *Y. lipolytica*[9]. The application of one-pot Golden Gate (GG) assembly enables efficient system for building several TUs on the vector bearing required HAs[23]. However, changing the HAs of a constructed integration cassette, usually delays the engineering progress, because it often involves the generation of a new cassette from scratch, or the use of less convenient cloning technique such as restriction-ligation or Gibson assembly. Being able to easily redirect a pre-assembled donor construct to an alternative integration locus would accelerate metabolic engineering approaches. Besides, it would enable integration of multiple copies of a TU across the genome and provide the opportunity to substitute one construction with another at a particular locus. We therefore hypothesise that an ideal CRISPR/Cas9-mediated integration system should possess the ability to exchange HAs on pre-assembled Cas9 donor constructs, using a simplified GG reaction.

**Re-encoding gRNA**. In contrast to other techniques, CRISPR/Cas9-mediated integration requires designing, assembling and transforming an additional construct expressing a variable gRNA sequence. A common and efficient approach combines both genetic elements, the Cas9 and the gRNA, on the same episomal helper, which ensures their transient expression after co-transformation with a donor[24]. To redirect the cutting and recombination event to a new position, only 20 bases of the gRNA sequence need to be changed. Several assembly techniques have been used with that purpose, including USER cloning[9], GG assembly[25], and cloning-free approaches[26]. In addition, a recent and efficient alternative method to modify a gRNA located on the *Escherichia coli* chromosome has been developed[27], which consists of using recombineering and a single 90-base oligonucleotide harbouring the specific 20 nucleotides in the middle of the sequence. Although recombineering is not efficient in eukaryotes, the required plasmid helper can be potentially assembled through a quick step in *E. coli*. We believe that implementing a rapid gRNA re-encoding method via recombineering could benefit CRISPR/Cas9-assisted integration systems.

A key element of any metabolic engineering project are promoters, which enable regulation of metabolic networks in order to redirect flux towards desired products[28]. Most yeast synthetic biology toolkits comprise a series of characterised promoters. However, in many non-conventional yeasts, NHEJ is strong and the use of marker-based technique to characterise promoters can be inaccurate as random integration can lead to multiple integration events[11,29,30]. At the same time, the characterisation of new promoters should ideally be made under a standardized genetic context, therefore as a single-copy integration into a defined genome position. This can be facilitated by CRISPR-mediated DSBs, known to improve integration over 1000 times[2,31]. Therefore, the incorporation of CRISPR-based promoter characterisation methods would be beneficial for many

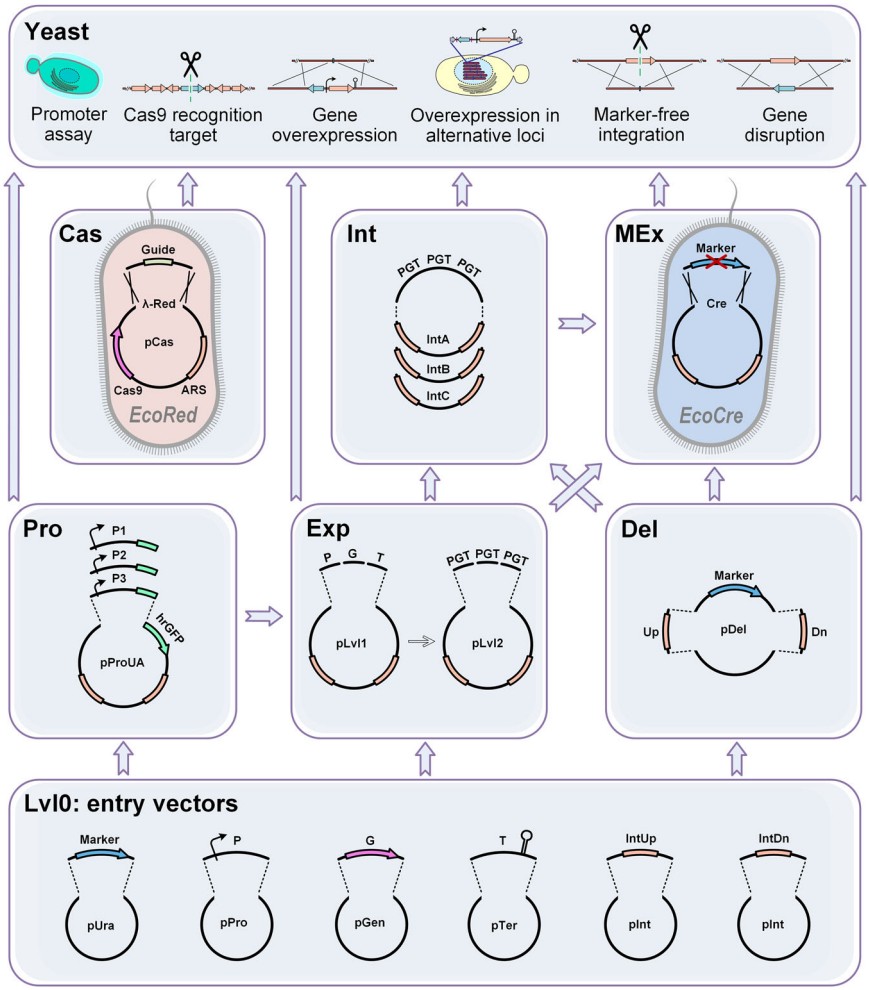

**Fig. 1 Modular toolkit structure.** The toolkit consists of seven modules for quick and easy assembly of integrative constructs and Cas9-helper plasmids. Lvl0 Module: single parts in entry vectors. Exp Module: assembly of overexpression constructs. Pro Module: assembly and screening of new promoters. Del Module: assembly of disruption constructs. Int Module: changing integration loci by homology arms exchange. MEx Module: assembly of marker-free constructs by selectable marker excision. Cas Module: redirection of Cas9-helper to new genome loci. Five of the modules - Pro, Del, Cas, Int, and MEx Modules - represent a new methodology that functionally extends previously used GG assembly systems. Assembly of Pro, Del and Int Modules are based on single GG reactions, while Cas and MEx Modules involve homologous and site-specific recombination taking place in the special *E. coli* strains. The arrows between modules indicate different orders in which they can be applied to enable variable genome engineering techniques as shown in the top panel, Yeast. Supplementary Table 8 is a list of the main plasmids of the toolkit.

yeast systems and would facilitate the easy and quick assays of large promoter libraries. In addition, it will allow a better comparison of promoter strengths between labs.

Here, using the industrial yeast *Y. lipolytica* as an example we have developed a modular metabolic engineering toolkit (*YaliCraft*) that combines well established gene editing and DNA assembly strategies with newly implemented methods to address the four concepts described above, which showed a high efficiency and versatility (Fig. 1).

*Y. lipolytica*, is an oleaginous yeast with high bioproduction capacity which is attracting significant attention from industry[32]. *Y. lipolytica*, as most non-conventional yeasts, shows a limited HR capacity and therefore the use of CRISPR/Cas9 mediated integration has proven effective[33–35]. Due to the relevance of this yeast, several modular toolkits have been developed for this organism[9, 36–41], although none of them directly addresses the issues described above.

We present here the *YaliCraft* toolkit, fully described in a detailed 44-page manual (Supplementary material), which is formed by seven modules that can be applied in different

combinations and enable the overexpression or disruption of genes, the redirection of gRNA and donor, the simultaneous assembly of donor with and without marker, as well as the screening of large libraries of promoters making them immediately available for gene overexpressions. To demonstrate the utility of the developed CRISPR/Cas9 system, we characterised the largest bundle of promoters ever described in *Y. lipolytica* and engineered a strain producing high titres of homogentisic acid (HGA) de novo from glucose.

## Results

**Modular architecture of the toolkit.** To unlock the full potential of CRISPR/Cas9 technology for metabolic engineering we developed a toolkit with a structure based on 7 individual modules which expands on previously well-known GG assembly systems[23,36,37,42]. We kept the syntaxis of previous Yarrowia toolkits to facilitate the exchange of compatible parts[36,37] and implemented a hierarchical structure based on the widely used yeast toolkit[23]. Each module comprises a specific type of

molecular operation modifying the final vector. Such operations are either an in vitro one-pot GG reaction or in vivo recombination in *E. coli*. Each step requires two days to prepare constructs for the next modification step or for yeast transformation (Supplementary Table 1). Some operations, e.g. GG assembly and marker excision, can be combined in a single step. The modules have been designed to be used in different orders and combinations that allows the maximum degree of freedom, while streamlining the number of steps required to obtain a desired construct (Fig. 1 or Supplementary Fig. 1).

The Lvl0 (level 0) Module is formed by a collection of individual genetic elements, which can be easily expanded. In the Exp (Expression) Module these parts can be combined in a variety of integrative overexpression vectors with the desired combinations of promoters, genes, terminators, HAs, and selectable markers. The Pro (Promoter) Module enables the characterisation of new promoters in yeast and it is fully compatible with the Exp Module. The Del (Deletion) Module allows the single-step assembly of marker-based constructions for gene disruptions. The Int (Integration) Module implements the fast redirection of pre-assembled overexpression constructs to alternative genome loci via HAs exchange. The MEx (Marker Excision) Module was designed to allow the generation of a marker-free version of any integration construct using a specific assembly host *EcoCre*. The separate Cas Module supports the simplified recombineering-based assembly of Cas9-helpers using the assembly host *EcoRed*. The obtained helpers are able to cut the yeast genome in any defined position enabling marker-free integration.

To verify the functionality and compatibility of these modules we have arranged an initial example toolkit for metabolic engineering of *Y. lipolytica*. The basic set comprises 147 plasmids, two *E. coli* and three *Y. lipolytica* strains (Supplementary Tables 8 and 9), which will be available through Addgene.

**(Lvl0 Module and Exp Module) Assembly of basic parts and overexpression constructs**. These two modules are the central part of the toolkit and consist of a modified three-level system adapted from the *S. cerevisiae* MoClo toolkit[23]. To permit the interchangeability of parts between labs, restriction sites (Supplementary Table 3) and overhangs (Supplementary Table 2) were designed to be compatible with previous *Y. lipolytica* GG toolkits (Supplementary Figs. 2–6)[36,37,42]. The basic set of Lvl0 plasmids contains 40 different parts, including promoters, genes, terminators, selectable markers, HAs, as well as vector backbones with *E. coli* fluorescent reporters.

The Exp Module is the only module that may include multiple assembly steps (Supplementary Figs. 8, 10–15). Depending on the strategy employed, an empty vector must be chosen with the preferred marker, integration locus (i.e. HAs), level and sublevel as explained below. Each empty Lvl1 can be used to assemble a single TU using Lvl0 plasmids with the promoter, gene, and terminator of interest. Each assembled Lvl1 cassette can be used to either overexpress a single gene or to assemble a Lvl2 plasmid containing multiple TUs. The sublevel of the Lvl1 plasmids (1, 2 and 3) determines the position of the TU in the subsequent Lvl2 assembly. The sublevel of the Lvl2 plasmid (2 and 3) indicates the number of TUs that may be assembled on it. The provided toolkit has the capacity to assemble of up to three TUs together, but this number may be further increased in the future if needed[36].

Each Lvl1 or Lvl2 plasmid contains specific 500-bp HAs allowing integration of the overexpression cassettes in a defined genome locus. Initially, 16 intergenic regions in *Y. lipolytica* W29 were selected (Supplementary Table 5), functionally characterised (Supplementary Fig. 27) and used for assembly of a further 80 empty Lvl1 and Lvl2 vectors of different sublevels. The loci on

each chromosome have been numbered and named accordingly as IntA1, IntA2, etc. An extra set of vectors has been supplied with Zeta sequences that can be used for random integration[43]. In order to keep a standard nomenclature, we designed a naming system that abbreviates all the required information for each vector (Supplementary Figs. 7, 9). For example, pE8US1.1 is an empty Lvl1.1 vector with HAs for integration at locus 8 on the chromosome E, containing a *URA3* and spectinomycin resistance marker.

**(Pro Module) Assembly of new promoters**. The Pro Module is formed by the unique empty vector pProUA-mScarlet. This vector contains *hrGFP* gene under the regulation of the inserted promoter (Supplementary Figs. 24 and 25), which permits fluorescence-based activity assays. Consequently, any promoter placed on this vector becomes immediately available for both assembly of TUs and integration in *Y. lipolytica* for functional characterization (Fig. 1). Using the promoter of *ALK1* in *Y. lipolytica* as an example, the observed assembly efficiency of pProUA-ALK1 plasmid was 100% (18/18) (Supplementary Fig. 29 and Supplementary Table 17).

**(Del Module) Assembly of gene disruption constructs**. The Del Module is based on a single vector designed for the one-pot assembly of pDel-series plasmids that allow gene disruptions. The empty vector pDelUK-RG contains the *URA3* cassette flanked by both *RFP* (*mScarlet*) and *sfGFP* genes designed for bacterial expression. During the GG assembly both reporters are substituted by PCR-amplified HAs flanking the gene that need to be removed (Supplementary Figs. 16 and 18). The correct clones can be visually screened by the absence of both fluorophores (Supplementary Fig. 17). As an example, pDelUK-AAT1 vector was assembled for the disruption of the *AAT1* gene. Among the GFP/RFP-negative colonies, 50% (9/18) shown the correct structure by restriction analysis (Supplementary Fig. 30).

**(Int Module) exchange of HAs between the constructs**. The toolkit was designed to allow switching between different HAs with already-assembled TUs. This feature was achieved by introducing two Type IIS AarI sites separating the vector backbone with HAs from the central section harbouring TUs. This allows reversible GG part exchange between empty and assembled overexpression vectors, which can be selected by altered antibiotic resistance and dropout of *sfGFP* (Supplementary Fig. 19). To demonstrate the efficiency of HA exchange we transferred three TUs from pZUA2.3-HPD1-ARO4-ARO7 (with Zeta integration flanks) to the empty vector pE8US1.1 for integration in IntE8 locus. 100% (8/8) of the transformants selected by spectinomycin resistance and GFP-negative phenotype comprised the plasmid pE8US-HPD1-ARO4-ARO7 (Supplementary Fig. 31).

In order to make the HAs from the Del Module available for the integration of overexpression constructs, the *URA3* marker on pDel-series was flanked by two AarI sites, which enable drop-out of this marker. Performing a GG reaction enables the irreversible transfer of TUs from Lvl1 and Lvl2 vectors into assembled pDel-series bearing HAs of a target gene (Supplementary Fig. 20). Using constructions pE8US-HPD1-ARO4-ARO7 and pDelUK-AAT1, we assembled the plasmid pDelUK-AAT1::HPD1-ARO4-ARO7 with efficiency 58.3% (7/12) (Supplementary Fig. 32).

**(MEx module) excision of the yeast selectable marker**. The MEx module permits a quickly switch from marker-free to marker-based integration without additional GG steps. This is achieved by the transformation of any required GG reaction mixture

(i.e. from Exp, Del or Int Modules) into two different *E. coli* strains. One of these strains is a regular transformation host (e.g. DH5alpha), while the other is the *EcoCre* strain, engineered to overexpress Cre recombinase (Section 1.6 of Supplementary material). Since the marker sequence is flanked by Lox66 and Lox71 sites[44] this part is instantly eliminated in *EcoCre* cells (Supplementary Fig. 21). Assembling both versions of plasmids in parallel allow us to immediately reverse to a marker-based approach in cases where marker-free integration is not efficient. The *URA3* excision using the *EcoCre* strain with the plasmids pC2US1.1-hrGFP, pD12US1.1-hrGFP, and pE8US1.1-hrGFP showed an efficiency of 100% (24/24) (Supplementary Fig. 33).

**(Cas Module) Re-encoding of gRNA on the Cas9-helper plasmid**. The Cas9-helper is an episomal vector that provides nourseothricin (Nat) resistance and expresses both the Cas9 and the guide (gRNA). This module enables the assembly of a new gRNA using a single 90-base oligonucleotide encoding the 20 bases required for the sequence recognition (Supplementary Figs. 22 and 23). This oligonucleotide is co-transformed with the empty helper pCasNA-RK in the recombineering *E. coli* strain *EcoRed*. Since pCasNA-RK contains a counter-selectable cassette (*rpsL-kanR*), desired recombinants can be isolated by streptomycin resistance. The assembly efficiency of Cas9-helpers was tested using three randomly generated 20-base recognition sequences, which resulted in the plasmids pCasNA-Rdm1, pCasNA-Rdm2 and pCasNA-Rdm3 (Supplementary Table 17). The percentage of correct clones was 79.4% (27/34) (Supplementary Fig. 34). The toolkit contains a set of pre-assembled Cas9-helpers for marker-free integration into 16 standard loci (Supplementary Table 8).

**Marker-free gene disruption and overexpression**. Efficiency of marker-free knockouts was assessed using *ARO8* and *ARO9* genes. A *Y. lipolytica* Ku70-mutant strain was co-transformed with different combinations of Cas9-helper and marker-free gene disruption cassette. Three alternative gRNA sequences were tested for each gene. The disruption efficiencies in separate experiments varied between 50% and 100%, while the average of the three was 77.8% (28/36) (Supplementary Fig. 35).

To check efficiency of marker-free integration of overexpression constructs the loci IntC2, IntD12, and IntE8 were selected. Accordingly, three marker-free Lvl1 plasmids harbouring the *hrGFP* gene under *TEF1* promoter were co-transformed with corresponding Cas9-helpers in prototrophic Ku70-mutant *Y. lipolytica* strain. Several fast-growing colonies, selected by Nat-resistance, from each transformation experiments were verified by colony PCR and assayed for green fluorescence. The efficiency of marker-free integration varied between 44.4% and 88.9%, with an average of 64% (32/50) (Supplementary Fig. 36).

**Promoter library characterisations**. To test the screening system, several promoter libraries were generated. As a source of promoters, we chose *Y. lipolytica* ribosomal genes encoding proteins of large (38 genes) and small (26 genes) subunits, as well as 29 other genes expected to be highly expressed (Section 8.1 of Supplementary material). Besides, a library of 43 hybrid promoters was designed combining the upstream regions of genes from several different yeast species, including *Candida hispaniensis*, *Kluyveromyces lactis*, *Kluyveromyces marxianus*, *Komagataella phaffii*, *Ogataea polymorpha*, *S. cerevisiae*, and *Y. lipolytica* (Section 8.2 of Supplementary material). All tested promoters were assembled as pPro-series plasmids and integrated in IntC2 locus of Ura⁻ Ku70-mutant strain using Cas9-helper and Nat-selection. Transformants were first verified by uracil prototrophy and then confirmed by PCR. Strong promoters were

visually screened by biomass fluorescence (Fig. 2, Supplementary Figs. 28 and 43). The relative strength of each promoter was measured using either plate reader or flow cytometry during exponential growth phase in two different media (Fig. 2, Supplementary Tables 12–15). We identified a variety of promoters with different strength that significantly expands the number of characterised promoters for *Y. lipolytica*. Interestingly, we identified 7 promoters stronger than TEF1, 5 hybrid and 2 ribosomal promoters.

**Metabolic pathway engineering**. To prove the utility of our enhanced CRISPR/Cas9 methodology we decided to create an HGA-producing *Y. lipolytica* through rational engineering. Under alkali conditions HGA is spontaneously oxidized and self-polymerises to form pyomelanin, which is an excellent constituent of natural sunscreens and cosmetics[45]. In addition, HGA is the biochemical precursor of two different families of high-value molecules, plastoquinols and tocopherols[46]. Despite its high commercial potential, the available technologies for the production of HGA and pyomelanin rely on the biotransformation of expensive aromatic amino acids[47]. Aromatics are biosynthesised through the shikimate pathway and starts from two intermediates of central metabolism, phosphoenolpyruvate and erythrose-4-phosphate. Tyrosine shares with HGA the common intermediate 4-hydroxyphenylpyruvate, a direct precursor for both molecules (Fig. 3a).

First, we selected several genes encoding putative aromatic aminotransferases as targets for engineering. This activity draws away carbon flow from HGA and leads to accumulation of tyrosine and phenylalanine. Furthermore, these two amino acids are involved in the feedback inhibition of several steps of the shikimate pathway[48, 49]. Most organisms possess two types of aromatic aminotransferases which show similarity either to the Aro8 protein of *S. cerevisiae* or TyrB of *E. coli*, which is closely related to Aat1 of *Aspergillus sp*[50,51]. In *Y. lipolytica*, we observed two paralogues of each type, named as *ARO8*, *ARO9*, and *AAT1*, *AAT2*, respectively. Using marker-free integration approach, we disrupted three (*ARO8*, *ARO9*, and *AAT2*) out of four putative aminotransferase genes in a Ku70-mutant and isolated strains with all possible combinations of these three deletions (Supplementary Fig. 37). However, marker-free disruption of the fourth aminotransferase gene (*AAT1*) in the triple mutant was not successful despite numerous attempts. Notable, the same deletion of *AAT1* gene worked well in a parental strain. This result suggested that the inactivation of the fourth gene adversely affected the viability of *Y. lipolytica*. In order to increase the selection pressure, we switched to a marker-based construct containing the *URA3* gene. In accordance with our assumptions, application of the auxotrophic marker allowed us to isolate a strain with all four gene disruptions, as well as other strains with combinations of these four deletions. The slow growth phenotype was always observed in transformation experiments combining deletions of *AAT1* and *AAT2* genes (Supplementary Figs. 38 and 39). Hence, these results suggest an epistatic interaction - also known as synthetic enhancement[18] - between these two genes.

Next, we selected three overexpression targets with the potential to enhance the de novo synthesis of HGA; *ARO4*, *ARO7* and *HPD1*. We chose the mutated *S. cerevisiae* genes *ARO4^{K229L}* and *ARO7^{G141S}* encoding feedback resistant enzymes which are known to enhance the metabolic fluxes through the shikimate pathway[52,53]. The third gene, *HPD1*, encodes a 4-hydroxyphenylpyruvate dioxygenase of *Y. lipolytica*[54]. All three genes were assembled with the strong *TEF1* promoter in the Lvl1 vectors and combined in a Lvl2 cassette. Attempts to integrate all three TUs together using a marker-free approach did not result in

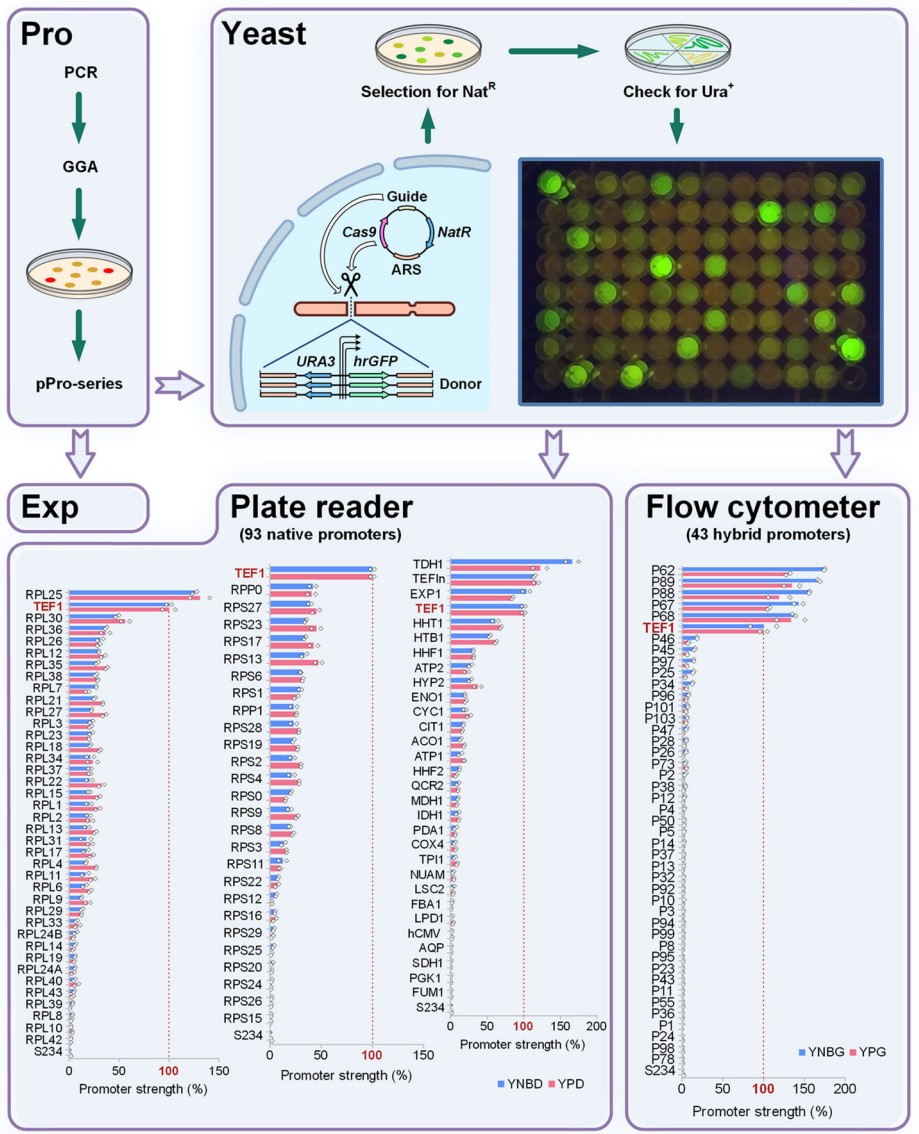

**Fig. 2 Promoter library screening using CRISPR/Cas9.** In the Pro Module an amplified promoter can be easily assembled in a single GG step using RFP dropout selection. Resultant pPro-series plasmids can be used for both TU assembly in the Exp Module and for promoter assay after integration into the yeast genome. Single-copy integration into the standard locus is facilitated by a co-transformed Cas9-helper plasmid. Transformants are first selected by Nat-resistance encoded by the episomal helper, followed by Ura$^+$ phenotype verification confirming integration of the construct. The 96-well plate pictured demonstrates fluorescence of the *Y. lipolytica* transformant library with 93 native promoters grown in YPD medium (Section 8.1 of Supplementary material). Two bottom panels contain bar charts with 93 native and 43 hybrid promoters that were assayed using plate reader and flow cytometer respectively. Data were blanked using the parent strain S234 (0%) and normalized by *TEF1* promoter activity (100%). For each promoter the GFP florescence was assayed in minimal (blue) and rich (red) media with either 2% glucose (YNBD, YPD) or 2% glycerol (YNBG and YPG) as the carbon source. Two biological replicates using independent transformants with the same construct are shown. Source data are provided as Supplementary Data 1.

the isolation of desired transformants regardless of the recipient strain used, e.g. with or without aminotransferase gene deletions. Transformation with all three genes separately using a marker-free approach let us overexpress *ARO7$^{G141S}$* and *HPD1* genes, while no colonies were isolated with *ARO4$^{K229L}$* (Section 7.1 of Supplementary material). At the same time, a control experiment using marker-based construction enabled *ARO4$^{K229L}$* overexpression, however correct clones always had smaller size in comparison with incorrect transformants (Supplementary Fig. 40). Due to the growth defects of both *ARO4$^{K229L}$* overexpression and Δ*aat1*Δ*aat2* double deletion, we decided to assemble these two modifications together using combined CRISPR/Cas9 and

marker-based selection. Both triple (Δ*aro8*Δ*aro9*Δ*aat2*) and quadruple (Δ*aro8*Δ*aro9*Δ*aat2*Δ*aat1*) mutant strains were co-transformed with the marker-based construct (pE8US-HPD1-ARO4-ARO7) and the corresponding Cas9-helper (pCasNA-IntE8). We first selected for the helper plasmid in the liquid medium with Nat, and then for the integrative construction on the solid medium without uracil. This method enabled the isolation of both triple and quadruple aminotransferase mutant derivatives with the overexpression of three required genes, designated as S946 and S948, respectively.

Finally, we decided to investigate and inactivate the degradation pathway of HGA. However, this pathway was yet unknown in *Y.*

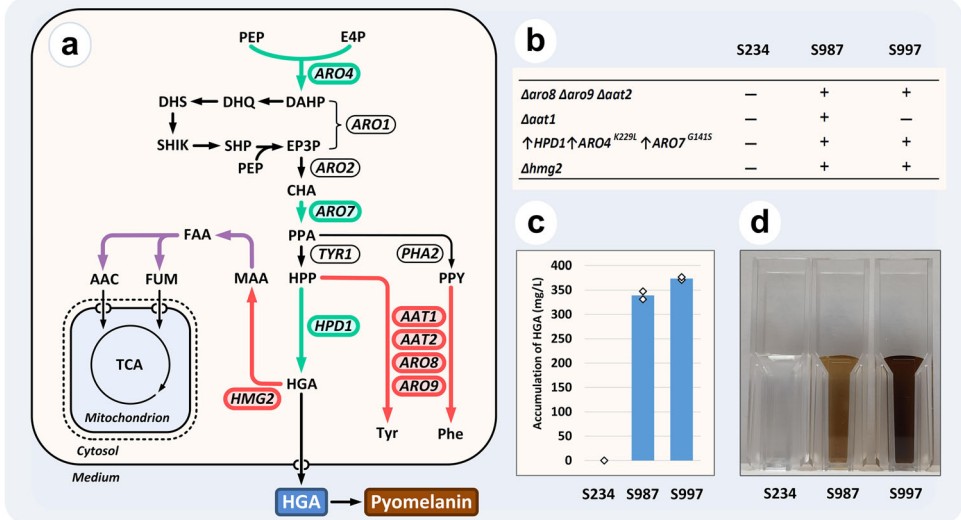

**Fig. 3 Validation of the toolkit for metabolic engineering. a** Schematic representation of the HGA producing pathway engineered in *Y. lipolytica*. Green arrows, overexpressed steps. Red arrows, inactivated reactions. Purple arrows, enzymatic reactions thought to be encoded by two other ORFs in the *Y. lipolytica* HGA degradation cluster. Genes encoding corresponding enzymatic steps are shown next to the reactions. Metabolites are shortened as follows: PEP phosphoenolpyruvate, E4P erythrose-4-phosphate, DAHP 3-deoxy-D-arabino-heptulosonate 7-phosphate, DHQ 3-dehydroquinate, DHS 3-dehydro-shikimate, SHIK shikimate, SHP shikimate-3-phosphate, EP3P 5-enolpyruvyl-shikimate-3-phosphate, CHA chorismate, PPA prephenate, HPP para-hydroxy-phenylpyruvate, PPY phenylpyruvate, MAA 4-maleyl-acetoacetate, FAA 4-fumaryl-acetoacetate, FUM fumarate, AAC acetoacetate, Tyr L-tyrosine, Phe L-phenylalanine. **b** Summary of modifications introduced in HGA producing strains. The symbols of delta and up arrow are used to indicate deletion and overexpression respectively of corresponding genes. **c** Accumulation of HGA by engineered strains after 14-day cultivation in YNB medium with 9% glucose. Two biological replicates are shown. Source data are provided as Supplementary Data 1. **d** Visible accumulation of pyomelanin by engineered strains in the cultural broth after 7-day cultivation in the YNB medium with 2% citrate.

*lipolytica*. In most organisms, including fungi, the HGA degradation pathway starts with the activity of homogentisate 1,2-dioxygenase[55]. Detailed analysis of *Y. lipolytica* W29 genome and resequencing of the selected region allowed us to identify an ORF missed during the previous whole genome analysis. We designated this ORF as *HMG2* and it encodes a protein which is highly similar to homogentisate 1,2-dioxygenase from other species (GenBank accession number MZ387986). This gene is the last unique sequence on chromosome D followed by repetitive elements. Interestingly, next to this gene we also identified two genes encoding putative fumarylacetoacetate hydrolase and glutathione S-transferase, which together with *HMG2* are suggested to be involved in HGA degradation in other fungi[56]. Furthermore, it is well documented that *Y. lipolytica* frequently produces mutants secreting a brown pigment[57,58], which could be pyomelanin formed from components of rich media. Due to the position of *HMG2* gene we anticipated that such mutants might be associated with spontaneous truncation of this telomeric region. Therefore, we decided to induce such a truncation artificially using a Cas9-helper that cut inside of *HMG2* gene. Indeed, transformation of a strain with wild type background induced intensive formation of brown pigment on rich media (Supplementary Fig. 42). Following this, we induced similar truncations in the strains S946 and S948, which led to the creation of S997 and S987 respectively (Fig. 3b). We found that both strains were able to synthesise HGA and pyomelanin de novo, on media with 9% glucose and 2% citrate as single carbon sources respectively (Fig. 3c, d). After 14 days of incubation in minimal medium with 9% glucose, the strain S997, a derivative of the triple aminotransferase mutant, was the best producer with 373.8 mg/L of HGA, while S987 produced 339.1 mg/L of HGA (Supplementary Table 11).

## Discussion

Future progress in metabolic engineering of living organisms will rely, to a great extent, on the availability of rapid, efficient DNA

manipulation technologies. Thanks to hierarchical and modular GG systems, the quick assembly of multiple TUs is no longer a major limiting factor—However, current engineering toolkits still show room for improvement to become more versatile and widespread, for example by further facilitating the iterative genomic integration of overexpression and deletion constructs.

For yeast metabolic engineering purposes, the application of CRISPR/Cas9 eases the consecutive integrations of DNA thanks to its marker-free capabilities. When compared with a system utilizing single integrative marker this technology accelerates strain construction by more than two folds (Fig. 4 and Supplementary Fig. 26). However, the construction of industrially relevant modifications, in many cases, still requires the robustness of marker-based selection. Surprisingly, even in our demonstrative metabolic engineering process we encountered toxic effects of both overexpression ($ARO4^{K229L}$) and gene disruption (combination of $\Delta aat1$ and $\Delta aat2$), which we were only able to engineer using marker-based constructs. These cases demonstrate that the capacity for switching back to a marker-based approach is advantageous for quickly engineering complex pathways. Here, we provide a solution (MEx Module) allowing the same DNA assembly to be used to produce both marker-free and marker-based constructs. Therefore, whenever a marker-base construction is required, this module can save several working days by skipping the need for de novo reassembly.

In addition, metabolic engineering using CRISPR/Cas9 requires frequent redirection of the donor DNA and the gRNA constructs towards alternative genomic loci. While that can be achieved using standard cloning techniques requiring amplification and/or gel-purification of DNA fragments, we have not been able to find in the literature an example where both HAs can be replaced in a single assembly step. Here, we demonstrate a system of one-pot HA exchange, which comprises a single GG reaction between two undigested plasmids (Int Module). This enables the redirection of complex donors to any available integration loci,

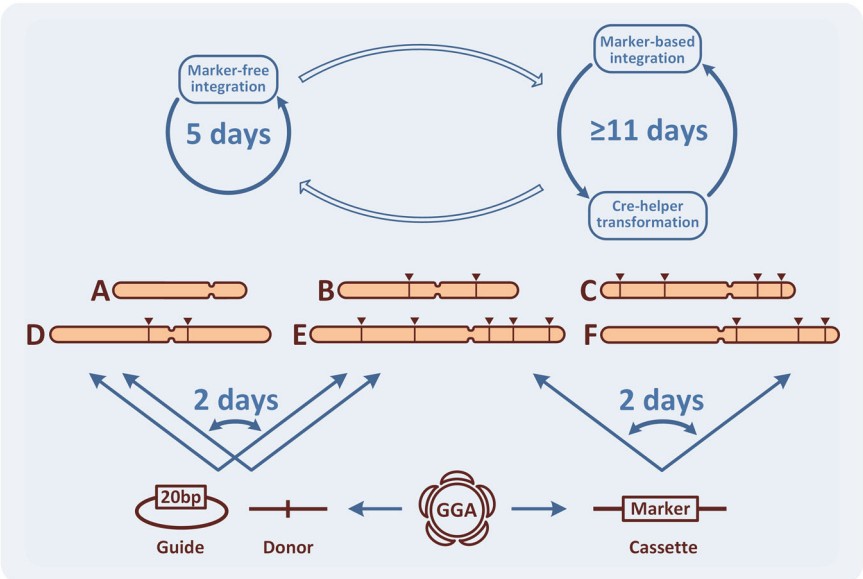

**Fig. 4 The principle of accelerated metabolic engineering using the CRISPR/Cas9 toolkit.** The toolkit allows frequent switching between marker-based and marker-free integration combining the advantages of both technologies. More robust marker-based integration requires at least 11 days, since it includes the marker recovery procedure. Application of the marker-free integration with CRISPR/Cas9 enables a single integration round in 5 days (image, top). Any single GG assembly reaction could be used for production of vectors with and without marker. At the same time, using the HA exchange system, any assembled integration cassette for overexpression, regardless of its complexity, can be redirected to alternative loci of the genome in 2 days. Moreover, both gRNA and donor can be redirected to alternative loci just in 2 days (image, bottom left).

including deletion targets, in only two days (Section 1.5 of Supplementary material).

Furthermore, although many techniques are now available for gRNA re-encoding on Cas9-helper plasmids, they greatly vary in the number and the complexity of molecular manipulations. We decided to bypass complex in vitro steps by using direct recombination of a single oligonucleotide on the empty helper inside the engineered bacterial assembly host (Cas Module). This shortens the time needed for assembly to the length of time required for bacterial transformation, which would be faster than any cloning-based techniques (Supplementary Table 4).

Another pivotal component of metabolic engineering projects is to achieve a tight control of gene expression. This requires the availability of a wide spectrum of well characterised promoters for individual host organisms. Using the unique properties of CRISPR/Cas9 to ensure precise integration of single copies, we arranged an efficient promoter screening system (Pro Module). A simple cloning step makes promoters immediately available for both assembly of TUs and in vivo testing in yeast.

In combination, the methodologies described above allow us to improve current CRISPR/Cas9 technologies and enhance their potential for yeast metabolic engineering. Using the industrial workhorse *Y. lipolytica* we constructed an initial toolkit with a new and versatile architecture. All procedures were designed to maximise speed and minimise complexity. The efficiency of all designed steps was estimated to be at least 50%, which makes the toolkit effective.

To assess the efficiency of our promoter screening system, we comparatively studied the expression of 137 promoters during growth in two different media. Among the 43 hybrid promoters tested we observed five which drove higher fluorescence than the widely used TEF1 promoter. All of those contained the proximal region of the *Y. lipolytica TEF1p*, and displayed higher strength than the non-truncated version. This may have resulted from the removal of a specific transcription regulator binding site repressing *TEF1* expression. If present, this site would lie in the distal region which was substituted in these five strong hybrid promoters. Among 94 natural promoters tested, two (*RPL25* and *TDH1*) also demonstrated expression levels higher than the previously known strongest wild type promoter *TEFin*, a variation of the *TEF1* promoter with the natural intron[59]. These new strong promoters will be of great interest for *Y. lipolytica* engineering. The rest of the library showed a large variety of expression levels (between 0.1% and 104% of *TEF1p*), which may be helpful for fine-tuning of metabolic pathways. This is the largest library ever tested in this organism, significantly expanding the total number of characterized promoters available in *Y. lipolytica*[60–62].

As a proof of concept of the applications of the toolkit in metabolic engineering, we constructed a *Y. lipolytica* strain producing 373.8 mg/L of HGA in minimal media. For this purpose, we combined previously known and several newly characterised targets for overexpression and deletion. Interestingly, we discovered a 'hidden' HGA degradation pathway, which is absent in two out of the five sequenced *Y. lipolytica* strains, WSH-Z06 and CLIB122. Although the co-linear gene cluster presents itself in evolutionarily distant mycelial fungi, we cannot assume horizontal gene transfer due to the low sequence similarity (Supplementary Fig. 41). On the contrary, the location near to the end of the chromosome suggested its spontaneous phylogenetic loss as the most likely event. Induced truncation of this telomere allowed us to obtain pyomelanin over-accumulating mutants with a characteristic brown phenotype. Spontaneous mutants with such phenotype are frequent in *Y. lipolytica* and lead to well-known problems in cheesemaking[63]. The discovery of this gene cluster and its location shed light on the genetic background of this phenomenon. While this manuscript was under preparation a study was published describing the overexpression of eight different genes in *Y. lipolytica*, resulting in the first example of HGA biosynthesis de novo with a final titre of 650 mg/L[54]. However, in that work, the largest increase in production was achieved in the final engineering step through the isolation of a unique clone producing 15-times more HGA than other transformants with the same construction. Notably, this clone was selected based on its intensive brown pigmentation, which we

hypothesise may be the result of a similar spontaneous telomere truncation to that described here.

In summary, our work presents an example of enhanced molecular toolkit tailored for CRISPR/Cas9-based metabolic engineering. We have proved its utility for both rapid strain construction and the characterisation of a large library of promoters. We anticipate that this toolkit may enable wider applications of this cutting-edge technology in strain engineering for industrial purposes. At the same time, the versatile structure of the toolkit enables extension of its capabilities, for instance to increase the number of integration loci, assemble more TUs together or include new modules such as those for CRISPR/Cas9 base editing, CRISPR activation or CRISPR inhibition, or CRISPR multiplexing. Additionally, the methodological solutions proven here in the *Y. lipolytica* model may be applied to other organisms, as well as to other fields of biological engineering.

## Methods

**Strains and media**. For *Escherichia coli* cultivation Luria Bertani media (VWR) was used. Yeast minimal growth medium was YNB (6,7 g/L yeast nitrogen base w/o amino acids, Sigma-Aldrich) supplemented with glucose (VWR, 101174Y) (YNBD), glycerol (VWR, 24388.320) (YNBG), or sodium citrate (VWR, 27833.237) at various concentrations as specified in the text. When required YNB was further supplemented with casamino acids (Thermofisher, #223050), uridine (Sigma-Aldrich, U3750), amino acids [L-Leucine (Sigma-Aldrich, L8000), L-Histidine (Sigma-Aldrich, H8000), L-Tyrosine(Sigma-Aldrich, T3754), L-Tryptophan (Sigma-Aldrich, T0254), L-Phenylalanine(Formedium, DOC0172), L-Aspartate (Sigma-Aldrich, 11195)], and phosphate buffer [Sodium hydrogen phosphate, Na2HPO4 (Alfa Aesar, 13437) and Sodium dihydrogen phosphate, NaH2PO4-2H2O(VWR, 28015.261)] as specified. Yeast rich growth medium was YPD (10 g/L yeast extract (Sigma-Aldrich, Y1625), 20 g/L peptone (Merck, 1.11931), and 20 g/L glucose) or YPG (10 g/L yeast extract, 20 g/L peptone, and 2% vol glycerol). The recipes of growth media are also provided in Supplementary Note 4. Antibiotics concentrations, including ampicillin (Sigma-Aldrich, A8351), kanamycin (Sigma-Aldrich, K1377), chloramphenicol (Acros Organics, 22792026), streptomycin (Sigma-Aldrich, S9137), spectinomycin (Sigma-Aldrich, S4014), nourseothricin (Fisher Scientific, AB-101-10ML), and hygromycin (Life Technologies, 10687010) are summarized in Supplementary Table 7. Hygromycin selection was used in maker recovery procedure as described in Section 2.5 of Supplementary material and Supplementary Table 6. Solid media were prepared by adding 20 g/L agar (Merck, 05039-500 G). *E. coli* XL1-Blue (Stratagene) and Turbo (NEB) were used for plasmid assembly and propagation. *Yarrowia lipolytica* strains used in this study are derived from uracil auxotroph (Δ*ura3*) and Ku70-mutant (Δ*ura3Δku70*) isolated from W29 (*MatA*, wild type; CLIB89) as was described before[64]. Genotypes of the strains that are components of the toolkit are summarized in Supplementary Table 9. These strains are available by request from the Russian State Collection of Industrial Microorganisms, VKPM.

**Plasmids and sequences**. All described plasmids were assembled using general cloning[65], Gibson assembly[66], Golden Gate assembly[67] or synthesized using commercial services (Twist Bioscience and Integrated DNA Technologies). The detailed protocols for assembly of DNA construct using YaliCraft toolkit are provided in Supplementary Note 1 and 3. The sequences of oligonucleotides used in this study are listed in Supplementary Tables 10 and 16. Plasmid sequences are provided in genbank format (Supplementary data_set). 147 plasmid components of the toolkit (Supplementary Table 8) will be available via the non-profit plasmid repository Addgene (##175597-175743).

**Recombineering**. *E. coli* strains *EcoRed* and *EcoCre* were derivatives of temperature-sensitive λ*cI*[857] lysogen of wild type strain MG1655. Both strains were constructed using recombineering[68,69] and contained the modified λ prophage for temperature-inducible expression of λ-Red genes (*gam, bet, exo*) or *cre* gene of phage P1, respectively. Strain *EcoRed* held deleted *cro-attR* region of the prophage sequence and contained *rpsL*[150] (Lys43Arg) mutation to enable counterselection with wild type *rpsL* gene[70]. Strain *EcoCre* in addition possessed a *N-attL* deletion in the prophage with *cre* gene inserted under the control of PL promoter. Sequences of the modified λ prophages corresponding to both strains are provided as Supplementary data_set.

**CRISPR/Cas9 genome engineering**. Both gRNA and Cas9 were expressed from the episomal pCasNA-series helper plasmid, which was modified pCRISPRyl vector[3]. The new vector contains the same autonomous replicative sequence, Cas9 expressing cassette, and regulatory elements for expression of sgRNA[71]. Codon-optimized *Cas9* gene from *Streptococcus pyogenes* contained C-terminal SV40 nuclear localization signal and was expressed from strong UAS1B8-TEF (136 bp) promoter[71]. The sgRNA

expression cassette is composed of four elements: hybrid RNA polymerase III promoter SCR1'-tRNAGly, the crRNA sequence (target specific 20 bp), followed by the tracrRNA sequence fused to poly-T polymerase III terminator sequence. The *LEU2* was replaced by nourseothricin resistance selectable marker. To increase the recognition efficiency, we removed 9-base of linker sequence from 5'-end of sgRNA[72]. The empty pCasNA-RK vector instead of target specific 20 bp contains counter-selectable cassette *rpsL-kanR*. For integration of overexpression or deletion construct, corresponding donor was co-transformed with Cas9-helper recognizing the same locus in *Y. lipolytica* genome (Supplementary Note 2).

**Promoter assay by flow cytometry and plate reader**. Activity of promoters in *Y. lipolytica* was measured by expression codon-optimized gene encoding humanized *Renilla* green fluorescent protein (*hrGFP*)[73]. Exponentially growing cells were analysed by flow cytometry using Attune NxT (Thermo Fisher Scientific). Fluorescence data were collected from 10,000 cells and analysed using FlowJo software (Ashland, OR). Alternatively, fluorescence was measured using a CLARIOstar Plus luminometer (BMG Labtech). Two different methods were used due to equipment availability in the different institutions. Similarly, two different media (glycerol and glucose) were used depending on the cryoprotectant agent used in each collection (glycerol or DMSO), to avoid undesired catabolic repression. Further characterisation experiments could test all promoters in the different medias and at different timepoints. Activity of promoters was blanked using autofluorescence of parent strain without GFP and normalized to the activity of *TEF1* promoter. Detailed promoter assay procedures by flow cytometry and plate reader described in Sections 2.6, 8.1, and 8.2 of Supplementary material.

**ORF identifications**. The genes coding for putative aromatic aminotransferases were identified by online tBLASTn search (http://blast.ncbi.nlm.nih.gov). As a result, four genes were assigned including *ARO8* (YALI1_E24922g), *ARO9* (YALI1_C06681g), *AAT1* (YALI1_B03198g), and *AAT2* (YALI1_F36966g). The ORF corresponding to *HPD1* (YALI1_B28454g) matched to a recently published sequence[54]. No locus tag had been assigned for *HMG2* gene during genome annotations due to apparent frameshift mutations (CP017556.1 and CP028451.1). The revised sequence of *HMG2* gene of *Y. lipolytica* W29 is available from GenBank under accession number MZ387986.

**Assaying strain productivity**. Homogentisic acid (HGA) concentration was estimated in the supernatant after biomass separation by UPLC/MS, using an Agilent 1290 Affinity chromatograph linked to an Agilent 6550 Q-ToF mass spectrometer. Separation was achieved using an Agilent Zorbax Eclipse Plus C18 column ($2.1 \times 50$ mm, 1.8 μm) and an acetonitrile gradient of 0% for 2 minutes then an increase to 98% over 0.5 minutes at a flow rate of 0.3 mL/min. Mass spectral data was acquired in negative ion mode from m/z 90 to 1000 at the rate of 3 spectra per second throughout the separation. 0.2 μL was injected from both sample wells and standard solutions prepared from commercially available HGA (H0751, Sigma-Aldrich).

**Statistics and reproducibility**. Experiments were done in duplicates or replicates as explained in each particular case. Average is shown in bar graph as well as individual data points.

**Reporting summary**. Further information on research design is available in the Nature Portfolio Reporting Summary linked to this article.

## Data availability

Source data are provided as Supplementary Data 1. Plasmid sequences are provided in genbank format as Supplementary data_set. Components of the toolkit will be available via the non-profit plasmid repository Addgene (##175597-175743). Any remaining information can be obtained from the corresponding author upon reasonable request.

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

## Acknowledgements

We thank Alice Boo for help with plate reader experiments and statistical analysis, Martin Todd for assistance with promoter library screening, Will Newell for proofreading of the manuscript, and David Bell for analytical support. We would also like to thank Eliza Atkinson, Diego Ruiz, Young-Kyoung Park, Razieh Rafieenia, Piotr Hapeta, Mohamed Almarei for providing valuable feedback on the Manual. This work was funded by Kaesler Nutrition GmbH, Biotechnology and Biological Sciences Research Council (BBSRC - BB/R01602X/1), BioMediCan Inc. and federal grant of Russian Federation according to agreement No. 075-15-2019-1659.

## Author contributions

T.V.Y. and R.L-A. conceived the Exp Module. T.V.Y. and E.Y.Y. conceived the Pro, Del, Cas, Int, and MEx Modules. T.V.Y. and E.Y.Y. developed a new methodology. R.L-A supervised the work. T.V.Y. and O.E.M. designed and assembled most plasmid parts and native promoters. H.D. designed the hybrid promoters. T.V.Y. and D.P. conducted the flow cytometry and wrote the manual. T.V.Y. and D.M.B. developed in vivo recombination techniques. T.V.Y. wrote the initial version of the manuscript. Subsequent versions were revised by R.L-A and H.D. H.D. and R.L-A. conceived the project supporting the research.

## Competing interests

Kaesler Nutrition GmbH declares that it has a competing interests in hybrid promoter sequences. All other authors declare no competing interests.
