## [Peer Review File · Communications Biology]

Reviewers' comments:

Reviewer #1 (Remarks to the Author):

The manuscript "A DNA assembly toolkit to unlock the CRISPR/Cas9 potential for metabolic engineering" describes the development of a new cloning toolkit based on Golden Gate Assembly. The authors identified three major drawbacks that existing toolkits lack, namely: Convenient return to marker-based integration, Redirection of donor DNA, and Re-encoding gRNA. The new YaliCraft toolkit is based on seven modules that are designed in a way that allows rapid and flexible genetic engineering of *Yarrowia lipolytica*. Additionally, the toolkit provides an impressive selection of characterized promoters and additional 16 characterized integration loci. As proof of functionality, a *Y. lipolytica* strain was engineered to produce homogentisic acid.

The key strengths of this toolkit are the wide range of functions and the interconnection and compatibility of the different modules. Additionally, it highlights the importance of marker-based integrations, first based on examples from the literature and then on the example of their own HGA production strain.

Furthermore, the YaliCraft Manuel is very well-written and extremely thorough. It contains easy-to-understand explanations and summarized protocols. Additionally, the illustrations and tables are of high quality and will be very helpful for newcomers to the field to grasp this complex topic. It will be an excellent guide for any future user of this toolkit and enable more labs to start genetic engineering projects in *Y. lipolytica*. Furthermore, while this toolkit is designed for *Y. lipolytica*, the same principles of DNA assembly can also be applied to the development of toolkits for other microbial cell factories.

The main areas of revision and key shortcomings of this manuscript are listed below. To summarize it lacks 1) a clear statement if it is a new toolkit or an addition of existing kits; 2) the engineering of multiple loci (deletion or integration) was not discussed; and 3) the details of the promoter characterization are not clear enough.

While this is not the first toolkit for *Y. lipolytica*, it is the newest version and extremely well thought through and well documented. New labs that are starting to use *Y. lipolytica* as a model organism, will most likely start using YaliCraft. It will be an essential engineering tool for many future studies. The concepts of YaliCraft can also be applied to other species. Overall, the potential contribution and impact on the field of metabolic engineering in yeast of this manuscript are very high.

Here is a list of comments that I think should/can be addressed to further improve the clarity and impact of the paper. I categorized my comments into three sections:

Major: things that I strongly think should be addressed

Minor: things that I think could be addressed to improve the manuscript but that are not vital

Text and typo: small improvements in the text that could improve clarity and structure and typos that I recognized

Major

1. It has been mentioned (line 143) that several modular toolkits for *Y. lipolytica* exist. They lack, however, the three aspects described in the introduction. From Figure 1 (which the introduction refers to) it is not clear which of the different modules are completely new, and which are already available in a similar form in existing toolboxes. It is not clearly stated if this is a new toolkit or an add-on to an existing one.

2. The toolkit is covering a wide range of applications. While the focus is on fast engineering it does not discuss multiplexing e.g. the deletion of multiple gene targets in a single transformation. I understand that the way the Cas module is designed this is not possible. Nevertheless, it should be

mentioned, and possible alternatives could be explained as to how a multi-deletion can still be achieved by this or other toolkits. For example, the EasyCloneYALI toolbox can target multiple loci in a single transformation, and the MoClo toolbox of *S. cerevisiae* recently got an expansion (10.1021/acssynbio.1c00408) which focuses on multiplexing (please don't feel obligated to include this reference though).

3. The characterized promoter libraries will be very valuable for the *Yarrowia* community. However, it is not clear to me why some of the promoters were tested in YNBD/YPD (plate reader) and others in YNBG/YPG (flow cytometer). Additionally, it is not explained why only one time point during the exponential growth was tested and how that time point was defined (was it at OD600 = 1?). For example, this study in *S. cerevisiae* (<https://doi.org/10.1093/nar/gkw1023>) has characterized four time points (4, 8, 24, and 48 hours) in three different media. However, their promoter library was significantly smaller (again, please don't feel obligated to include this reference either).

Minor

Main manuscript

4. In connection to the 3.: I fully understand that re-testing all promoters in different media at different time points would be a lot of work. However, maybe consider if you could test some of the most interesting promoters (those that have even higher expression than TEF1) could be further characterized to ensure that their activity is strong throughout the whole exponential phase and maybe even thereafter, or discuss this limitation in the discussion section.

5. It is not clear why two different instruments were used for the promoter characterization. Was it simply the availability of the instrument to parts of the team or was there a deeper scientific reason behind it? A short comment somewhere could clarify that.

6. The manual names the toolkit "YaliCraft", which I think is a great name that will help to recognize this toolkit. However, the name is not included in the main manuscript at all. You might even consider including it in the title of the paper. (This will also help to address my first comment.)

7. As a toolkit, one of its goals is the usage of many other labs. It is mentioned in the methods that the plasmids are available at Addgene, but I think it could be mentioned before already, that everything is publicly available, to promote it even further. Maybe around lines 177-179?

8. The toolkit includes 16 characterized standard integration loci. Initially, 50 different loci were selected but only 16 randomly selected loci were further characterized. Although they are not further characterized, it might be nice to supply the location of the remaining 34 loci, in case other researchers want to explore those additional loci to extend YaliCraft.

9. For the 16 integration loci, the authors showed the integration efficiency and the relative expression level of those loci. I would be curious if the integration of a TU at the different loci unintentionally influences the growth of the strain. Are there any growth curves or final ODs available that could give an idea if the integration interferes with the native metabolism?

10. It is not clear how the promoter region was defined for the natural promoter libraries. For the hybrid promoter library, it is set to 800 bp (Supplementary line 424) but this information is not clear for the natural promoters.

11. The HGA cultivation is not described in enough detail in the methods or caption of Figure 3. It is not clear if the cultivation was done in cultivation tubes, shake flasks, or bioreactors, which volume, temperature or aeration was used etc.

12. Additionally, I was wondering why a 14-day cultivation was chosen for the HGA production. Do you have additional information about the production at earlier time points? Furthermore, what is the growth of the strain compared to the wt strain S234 (OD or biomass)?

13. In the methods: for some of the chemicals the provider is stated, while it is missing for others. From my own experience I know it is not the most stimulating task to do, but adding the provider of the used chemicals can improve reproducibility, especially if more complex compounds are used (e.g. casamino acids line 1038 of the manual).

Figures

First of all, I'd like to point out that all figures are very well prepared and aesthetically designed. It shows that quite some time was put into their creation which future readers of this paper will appreciate. However, I have some comments on some of them that might improve them even further.

Fig1:

14. Is that the best figure for the beginning? Most of the modules are not described in the introduction and the three concepts explained are not marked in the figure. Additionally, in lines 135-138, you mention that this kit is based on "well established gene editing and DNA assembly strategies with newly implemented methods to address the four concepts". From the figure, it is not clear which parts are completely new and which are mainly based on existing tools.

Fig2:

15. The x-axis label and unit are missing (it is getting clear out of the context of the text and figure caption, but it should also be marked in the plot). I think "promoter strength (%)" would be sufficient and maybe adding a thin vertical line to TEF1 or highlighting TEF1 in bold or with colour could make the normalization clearer.

Fig3:

16. In subfigure c) the title should be the y-axis title

Manual

17. "1.6 MEx module": The marker excision is close to 100%, but it still needs to be verified. I assume this is done with a test digest that also confirms the correct assembly of the remaining plasmid? If so, it could be mentioned in one sentence to make it easier to understand for new users.

Text and typos

In general, the manuscript is well prepared and in a document of this size, one will always find another typo somewhere. These are not numbered, just address those that you agree with. For all documents, I'd recommend checking if space was used between numbers and units. Some sections of the manuscript and manual don't use the space.

Main manuscript

Line 128 – 131: The explanation of how CRISPR/Cas9 works feels a bit out of place for me. Latest since the "Convenient return to marker-based integration." section the reader should know the advantages and functions of CRISPR/Cas9. I would recommend either moving the two sentences further up or removing them.

Lines 135-138 feel out of place. During my first reading, I expected this to be the closing section of the introduction. It is also very similar to line 145ff.

Line 251: "a single step" should be "in parallel". Until I read this section I thought the toolkit allows the inclusion of a marker into an existing marker-free plasmid. But as I understand now, it is a clever way of assembling both versions (marker-based/free) in parallel from the same GG reaction. That was not clear enough for me here. It is very clear in the discussion though.

Typos

Main manuscript

88: "the" process

99: "the" application ... "an" efficient system

150: "a" strain

297: "TEF" is missing "1"

433: "ever" needs to be moved by one word

Supplementary:

366: 16th and 24th should be just "h"

Manual:

Cover: update the year to 2023

Finally, you made it through! Although I collected quite a long list of comments I would like to point out once again, that this is a great toolkit. The overall preparation of the manuscript and manual reflect the hard work that was put into this project. I'm sure this toolkit will find many users that will appreciate your commitment!

I hope my comments will help to further improve your manuscript.

OK

Reviewer #2 (Remarks to the Author):

The manuscript from Yuzbasheva and colleagues titled "A DNA assembly toolkit to unlock the CRISPR/Cas9 potential for metabolic engineering" describes a diligent work for systematization of a series of 147 plasmids comprising a toolkit for genetic engineering the yeast *Yarrowia lipolytica*. The level of organization of this work is quite impressive. The manuscript is well written and have the support of a Manual that is an important guide to use and understand the toolkit. In addition, practical applications illustrate the strength of their toolkit by the characterization of 137 promoters and the engineering of homogentisic acid production in *Y. lipolytica*. I believe, the present work is very interesting for those researchers striving to improve their knowledge on DNA recombinant techniques, and certainly will provide a good support for others specifically interested in *Yarrowia*. Therefore, I do support the manuscript publication, and only have minor comments as suggestions for the authors.

In my perception, the abstract is not quite direct about the essence of the work, which is the assembly of a formidable toolkit of 147 plasmids to support the metabolic engineering of *Y. lipolytica*. It is Ok to emphasize aimed improvements, but those come from using the plasmid toolkit, comprising seven specific modules, each one having a functionality. I also think it is important to mention early on that the toolkit is available through the Addgene repository.

I think of some tiny suggestions to improve the understanding of the system described in the Figure 1. A simple label such as "Level 0 (Lv0): entry vectors" could make clearer that the basal plasmids are the assembly pieces for the upper-level plasmids. When possible, it would be interesting to write a number describing how many plasmids of each kind (pPro, pTer, pInt, etc...) are available as part of the toolkit. I think that the explanations of modules present in the Manual as captions of Fig S1, could be incorporated in the Figure 1 caption, on the main manuscript. Probably, a reference to the Table S8 in the Figure 1 caption is also important.

Apart from those minor points, the manuscript seems robust to me, and I believe the readers can learn from it. I congratulate the authors for the good work.

Reviewer #3 (Remarks to the Author):

Technologies based on CRISPR/Cas9 have been developed to facilitate microbial cell engineering, even though they still have some limitations and do not always allow simple and fast genome editing. In their manuscript, the Authors have identified three of those limitations and proposed a corresponding

engineering solution. The presented YaliCraft toolkit is based on modules that extend the versatile Golden-Gate assembly system already available in *Y. lipolytica*. Each module comprises a specific type of molecular operation modifying the final vector, namely assembly of new promoters (Pro Module), assembly of gene disruption construct (DelModule), the possibility for quick exchange of homologous arms (HA) used for integration (Int Module), a module allowing the switch from marker free to marker-based integration (MEx Module) and a Cas Module allowing an easy and fast re-encoding of gRNA in Cas9-helper plasmid. As a proof of concept, the Authors have characterized a set of 137 promoters (Pro Module) and engineered *Y. lipolytica* either by gene disruption or overexpression to produce homogentisic acid. Beside the manuscript, the Authors propose a leaflet of 40 pages detailing protocols on how to use the YaliCraft toolkit. The manuscript and the YaliCraft manual are well written and structured. In its concept, this extended toolkit is compatible with those already available for *Y. lipolytica* engineering. This toolkit will be helpful for the *Yarrowia* community. I do not have specific comments on the manuscript.

Reviewers' comments:

Reviewer #1 (Remarks to the Author):

The manuscript “A DNA assembly toolkit to unlock the CRISPR/Cas9 potential for metabolic engineering” describes the development of a new cloning toolkit based on Golden Gate Assembly. The authors identified three major drawbacks that existing toolkits lack, namely: Convenient return to marker-based integration, Redirection of donor DNA, and Re-encoding gRNA. The new YaliCraft toolkit is based on seven modules that are designed in a way that allows rapid and flexible genetic engineering of *Yarrowia lipolytica*. Additionally, the toolkit provides an impressive selection of characterized promoters and additional 16 characterized integration loci.

As proof of functionality, a *Y. lipolytica* strain was engineered to produce homogentisic acid.

The key strengths of this toolkit are the wide range of functions and the interconnection and compatibility of the different modules. Additionally, it highlights the importance of marker-based integrations, first based on examples from the literature and then on the example of their own HGA production strain.

Furthermore, the YaliCraft Manual is very well-written and extremely thorough. It contains easy-to-understand explanations and summarized protocols. Additionally, the illustrations and tables are of high quality and will be very helpful for newcomers to the field to grasp this complex topic. It will be an excellent guide for any future user of this toolkit and enable more labs to start genetic engineering projects in *Y. lipolytica*. Furthermore, while this toolkit is designed for *Y. lipolytica*, the same principles of DNA assembly can also be applied to the development of toolkits for other microbial cell factories.

We would like to thank reviewer 1 for the positive comments and their enthusiasm about our toolkit.

The main areas of revision and key shortcomings of this manuscript are listed below. To summarize it lacks 1) a clear statement if it is a new toolkit or an addition of existing kits; 2) the engineering of multiple loci (deletion or integration) was not discussed; and 3) the details of the promoter characterization are not clear enough.

Thanks for pointing out how to improve the manuscript. We explain how we address these 3 aspects in the point-by-point answer below.

While this is not the first toolkit for *Y. lipolytica*, it is the newest version and extremely well thought through and well documented. New labs that are starting to use *Y. lipolytica* as a model organism, will most likely start using YaliCraft. It will be an essential engineering tool for many future studies.

The concepts of YaliCraft can also be applied to other species. Overall, the potential contribution and impact on the field of metabolic engineering in yeast of this manuscript are very high.

Thanks again for the very nice comments.

Here is a list of comments that I think should/can be addressed to further improve the clarity and impact of the paper. I categorized my comments into three sections:

Major: things that I strongly think should be addressed

Minor: things that I think could be addressed to improve the manuscript but that are not vital

Text and typo: small improvements in the text that could improve clarity and structure and typos that I recognized

Major

1. It has been mentioned (line 143) that several modular toolkits for *Y. lipolytica* exist. They lack, however, the three aspects described in the introduction. From Figure 1 (which the introduction refers to) it is not clear which of the different modules are completely new, and which are already available in a similar form in existing toolboxes. It is not clearly stated if this is a new toolkit or an add-on to an existing one.

Thanks for pointing that this aspect needs clarity.

This toolkit is new, except for the fact that it is build using the syntaxis (specific overhangs) of the first Golden Gate toolkit published in *Yarrowia* (<https://pubmed.ncbi.nlm.nih.gov/28217858/>). We decided to keep that syntaxis to improve transferability of parts between labs and keep compatibility with existing parts.

Some aspects, such as level 0 and the Exp module have been created based on the design of other popular toolkits, in particular the widely used *S. cerevisiae* YTK (<https://pubs.acs.org/doi/10.1021/sb500366v>).

We have now highlighted this in the beginning of the result section, which reads as follows:

“we developed a toolkit with a new structure, based on 7 individual modules which expands on previously well-known GG assembly systems^{23,37,38,43}. We kept the syntaxis of previous Yarrowia toolkits to facilitate the exchange of compatible parts^{37,38} and implemented a hierarchical structure based on the widely used yeast toolkit²³.”

2. The toolkit is covering a wide range of applications. While the focus is on fast engineering it does not discuss multiplexing e.g. the deletion of multiple gene targets in a single transformation. I understand that the way the Cas module is designed this is not possible. Nevertheless, it should be mentioned, and possible alternatives could be explained as to how a multi-deletion can still be achieved by this or other toolkits. For example, the EasyCloneYALI toolbox can target multiple loci in a single transformation, and the MoClo toolbox of *S. cerevisiae* recently got an expansion (10.1021/acssynbio.1c00408) which focuses on multiplexing (please don't feel obligated to include this reference though).

This is a great point. We have now added to the discussion on the potential extensions of the toolkit to include the compatibility with multiplexing strategies.

The extended text now reads:

“At the same time, the versatile structure of the toolkit enables extension of its capabilities, for instance to increase the number of integration loci, assemble more TUs together or include new modules such as those for CRISPR/Cas9 base editing, CRISPR activation or CRISPR inhibition, or CRISPR multiplexing.”

3. The characterized promoter libraries will be very valuable for the *Yarrowia* community. However, it is not clear to me why some of the promoters were tested in YNBD/YPD (plate reader) and others in YNBG/YPG (flow cytometer). Additionally, it is not explained why only one time point during the exponential growth was tested and how that time point was defined (was it at $OD_{600} = 1$?). For example, this study in *S. cerevisiae* (<https://doi.org/10.1093/nar/gkw1023>) has characterized four time points (4, 8, 24, and 48 hours) in three different media. However, their promoter library was significantly smaller (again, please don't feel obligated to include this reference either).

Thanks for recognizing the usefulness of the promoter library for the *Yarrowia* community. The reason why one set of promoters was characterized in plate reader and the other in flow cytometer was because the first set was tested in London and the second in Moscow and the available equipment was different. The protocols for each of them was also slightly different and that is why we used different media. In London, the cryovials in the culture collection were done with glycerol and to avoid traces of glycerol in a glucose-based media, we kept the media with glycerol but in Moscow, the cryoprotectant used was DMSO, allowing the use of glucose in the media. We have now clarified this in the methods section.

As suggested by the reviewers, due to the size of the library we were only able to take one time point. The selected point/conditions (P13 L364-368 Supplementary Experiments) were chosen based on a preliminary experiment where we screened multiple strains and identified the average mid-exponential phase (P17 L442-446 Supplementary Experiments). We have also clarified this now in the methods section.

The new text reads:

“Two different methods were used due to equipment availability in the different institutions. Similarly, two different media (glycerol and glucose) were used depending on the cryoprotectant agent used in each collection (glycerol or DMSO), to avoid undesired catabolic repression.”

Minor

Main manuscript

4. In connection to the 3.: I fully understand that re-testing all promoters in different media at different time points would be a lot of work. However, maybe consider if you could test some of the most interesting promoters (those that have even higher expression than TEF1) could be further characterized to ensure that their activity is strong throughout the whole exponential phase and maybe even thereafter, or discuss this limitation in the discussion section.

As suggested by the reviewer, this would require significant efforts and we have decided to discuss this limitation.

The new sentence now says:

“Further characterisation experiments could test all promoters in the different medias and at different timepoints.”

5. It is not clear why two different instruments were used for the promoter characterization. Was it simply the availability of the instrument to parts of the team or was there a deeper scientific reason behind it? A short comment somewhere could clarify that.

One of the experiments was done in the UK while the other was done in Russia, so the main reason is the availability of equipment. As discussed above, we have now clarified this in the methods section.

“Two different methods were used due to equipment availability in the different institutions.”

6. The manual names the toolkit “YaliCraft”, which I think is a great name that will help to recognize this toolkit. However, the name is not included in the main manuscript at all. You might even consider including it in the title of the paper. (This will also help to address my first comment.)

Thanks for the suggestion, we have now introduced the name in the abstract and other sections of the manuscript.

“As a proof of concept, we developed the YaliCraft toolkit for Yarrowia lipolytica, which is composed of a basic set of 147 plasmids and 7 modules with different purposes.”

“Here, using the industrial yeast Y. lipolytica as an example we have developed a modular metabolic engineering toolkit (YaliCraft) that combines well established gene editing and DNA assembly strategies with newly implemented methods to address the four concepts described above, which showed a high efficiency and versatility (Fig. 1).”

“We present here the YaliCraft toolkit, fully described in a detailed 40-page manual (Supplementary Manual), which is formed by seven modules that can be applied in different combinations and enable the overexpression or disruption of genes, the redirection of gRNA and donor, the simultaneous assembly of donor with and without marker, as well as the screening of large libraries of promoters making them immediately available for gene overexpressions.”

7. As a toolkit, one of its goals is the usage of many other labs. It is mentioned in the methods that the plasmids are available at Addgene, but I think it could be mentioned before already, that everything is publicly available, to promote it even further. Maybe around lines 177-179?

We have now included this earlier in the text as suggested.

“The basic set comprises 147 plasmids, two E. coli and three Y. lipolytica strains (Tables S8 and S9 of Supplementary Manual), which are available through Addgene.”

8. The toolkit includes 16 characterized standard integration loci. Initially, 50 different loci were selected but only 16 randomly selected loci were further characterized. Although they are not further characterized, it might be nice to supply the location of the remaining 34 loci, in case other researchers want to explore those additional loci to extend YaliCraft.

This is a valuable point. Since these additional loci have not been characterized, there is no real value in adding them in this work, so we have now removed them and only described the 16 characterised ones.

9. For the 16 integration loci, the authors showed the integration efficiency and the relative expression level of those loci. I would be curious if the integration of a TU at the different loci unintentionally influences the growth of the strain. Are there any growth curves or final ODs available that could give an idea if the integration interferes with the native metabolism?

We did not observe large differences in OD600 when we characterised these integration sites (Fig. S27) – all strains were able to produce OD600 values between 0.8 and 1.1 after 16 hours of cultivation at 600 rpm. In addition, the integration sites were selected to be located in intergenic spaces and distant from promoters and terminator, and it is generally expected that they do not affect the viability.

10. It is not clear how the promoter region was defined for the natural promoter libraries. For the hybrid promoter library, it is set to 800 bp (Supplementary line 424) but this information is not clear for the natural promoters.

We selected 800bp as the promoter region for all promoters. For native promoters, if the upstream gene was closer than 800bp, then we reduced the length accordingly. All sequences used are provided as the supplementary GenBank files.

11. The HGA cultivation is not described in enough detail in the methods or caption of Figure 3. It is not clear if the cultivation was done in cultivation tubes, shake flasks, or bioreactors, which volume, temperature or aeration was used etc.

The reviewer is correct and the HGA experiment is not described in detail in the methods section. However, we included a detailed description in Supplementary Experiments (P10 L245-250). While revisiting it, we have noticed that we did not mention that the incubation was done in 24-deepwell microplates (EnzyScreen, CR1424). We have now included this information and mentioned in the methods that this experiment is described in the Supplementary Experiments.

The text says:

“To check HGA production, YNB media with 9% glucose was inoculated with each strain to an initial OD600 of 0.1 in 2.5mL of culture using 24-deepwell microplates (Enzyscreen, CR1424). Three strains were analysed, including both final strains (S997 and S987) and the wild type parent (S234) as the negative control. Cultivation was performed at high aeration rate (300 rpm) at 30° C. After 14 days biomass was separated and HGA concentration was measured in the supernatant by UPLC/MS. The accumulation of HGA by the two engineered strains is summarized in Table S11.”

12. Additionally, I was wondering why a 14-day cultivation was chosen for the HGA production. Do you have additional information about the production at earlier time points? Furthermore, what is the growth of the strain compared to the wt strain S234 (OD or biomass)?

We used 14-days because one of the strains (S987) showed very slow growth (the growth defect was recovered in S997) and we wanted all strains to reach a decent OD before quantifying production. Since this experiment was only a proof of concept and not an optimization with a bioproduction goal, we only checked the final time point.

13. In the methods: for some of the chemicals the provider is stated, while it is missing for others. From my own experience I know it is not the most stimulating task to do, but adding the provider of the used chemicals can improve reproducibility, especially if more complex compounds are used (e.g. casamino acids line 1038 of the manual).

This is great point. We have now added in the methods section of the paper the provider of each reagent and the catalogue number, the first time they appear. They are also listed here:

Reagent	Supplier
Casamino acids	(Thermofisher, #223050)
Glucose	(VWR, 101174Y)
Glycerol	(VWR, 24388.320)
tri-sodium citrate dihydrate	(VWR, 27833.237)
Uracil	(Sigma-Aldrich, U0750)
Uridine	(Sigma-Aldrich, U3750)
L-Leucine	(Sigma-Aldrich, L8000)
L-Histidine	(Sigma-Aldrich, H8000)
L-Tyrosine	(Sigma-Aldrich, T3754)
L-Tryptophan	(Sigma-Aldrich, T0254)
L-Phenylalanine	(Formedium, DOC0172)
L-Aspartate	(Sigma-Aldrich, 11195)
Sodium hydrogen phosphate, Na ₂ HPO ₄	(Alfa Aesar, 13437)
Sodium dihydrogen phosphate, NaH ₂ PO ₄ ·2H ₂ O	(VWR, 28015.261)
yeast extract	(Sigma-Aldrich, Y1625)
peptone	(Merck, 1.11931)
agar	(Merck, 05039-500G)
ampicillin	(Sigma-Aldrich, A8351)
kanamycin	(Sigma-Aldrich, K1377)
chloramphenicol	(Acros Organics, 22792026)
streptomycin	(Sigma-Aldrich, S9137)
spectinomycin	(Sigma-Aldrich, S4014)
nourseothricin	(Fisher Scientific, AB-101-10ML)
hygromycin	(Life Technologies, 10687010)

Figures

First of all, I'd like to point out that all figures are very well prepared and aesthetically designed. It shows that quite some time was put into their creation which future readers of this paper will appreciate. However, I have some comments on some of them that might improve them even further.

Fig1:

14. Is that the best figure for the beginning? Most of the modules are not described in the introduction and the three concepts explained are not marked in the figure. Additionally, in lines 135-138, you mention that this kit is based on

“well established gene editing and DNA assembly strategies with newly implemented methods to address the four concepts”. From the figure, it is not clear which parts are completely new and which are mainly based on existing tools.

Thanks for the comment. This has been addressed while replying to “Major comment 1”, and now the text contains additional information about which aspects of the toolkit are new. In addition, figure 1 has been slightly modified according to Reviewer 2 for further clarity on module 0 and its link to previous toolkits.

Fig2:

15. The x-axis label and unit are missing (it is getting clear out of the context of the text and figure caption, but it should also be marked in the plot). I think “promoter strength (%)” would be sufficient and maybe adding a thin vertical line to TEF1 or highlighting TEF1 in bold or with colour could make the normalization clearer.

We have now modified the figure as suggested.

Fig3:

16. In subfigure c) the title should be the y-axis title

This has been modified.

Manual

17. “1.6 MEx module”: The marker excision is close to 100%, but it still needs to be verified. I assume this is done with a test digest that also confirms the correct assembly of the remaining plasmid? If so, it could be mentioned in one sentence to make it easier to understand for new users.

The description of the verification is described in detail in Supplementary Experiments (P3 L61-74). Since we would like the Supplementary Manual to be self-sufficient document, we have not added a reference to this section there, but we expanded the description in the Manual that says: “*Efficiency of the marker excision was 100% as verified by restriction analysis*”.

Text and typos

In general, the manuscript is well prepared and in a document of this size, one will always find another typo somewhere. These are not numbered, just address those that you agree with.

For all documents, I’d recommend checking if space was used between numbers and units. Some sections of the manuscript and manual don’t use the space.

We have now checked the spacing for consistency.

Main manuscript

Line 128 – 131: The explanation of how CRISPR/Cas9 works feels a bit out of place for me. Latest since the “Convenient return to marker-based integration.” section the reader should know the advantages and functions of CRISPR/Cas9. I would recommend either moving the two sentences further up or removing them.

We have decided to remove that part as suggested and modified the previous sentence slightly.

The new text reads as:

“This can be facilitated by CRISPR-mediated DSBs, known to improve integration over 1,000 times^{2,32}.”

Lines 135-138 feel out of place. During my first reading, I expected this to be the closing section of the introduction. It is also very similar to line 145ff.

We understand the point by reviewer 1. However, in this case, the first sentence is just introducing the overall objective of this work and indicating that we chose *Yarrowia* as an example. The next paragraph is to introduce *Yarrowia* and its importance to the readers and the final one, goes in more detail into the specific toolkit that has been created for *Yarrowia* and the cases of use developed in this work. We therefore believe that we should keep both sentences if the editorial team agrees.

Line 251: “a single step” should be “in parallel”. Until I read this section I thought the toolkit allows the inclusion of a marker into an existing marker-free plasmid. But as I understand now, it is a clever way of assembling both versions (marker-based/free) in parallel from the same GG reaction. That was not clear enough for me here. It is very clear in the discussion though.

Thanks for pointing this out. We have now clarified this in the text by indicating “in parallel” as suggested.

Typos

Main manuscript
88: “the” process
99: “the” application ... “an” efficient system
150: “a” strain
297: “TEF” is missing “1”
433: “ever” needs to be moved by one word

These typos have been corrected.

Supplementary:

366: 16th and 24th should be just “h”

Done

Manual:

Cover: update the year to 2023

Done

Finally, you made it through! Although I collected quite a long list of comments I would like to point out once again, that this is a great toolkit. The overall preparation of the manuscript and manual reflect the hard work that was put into this project. I'm sure this toolkit will find many users that will appreciate your commitment!

I hope my comments will help to further improve your manuscript.

OK

We would like to thank once again the valuable comments and the positive and constructive tone of reviewer 1.

Reviewer #2 (Remarks to the Author):

The manuscript from Yuzbasheva and colleagues titled “A DNA assembly toolkit to unlock the CRISPR/Cas9 potential for metabolic engineering” describes a diligent work for systematization of a series of 147 plasmids comprising a toolkit for genetic engineering the yeast *Yarrowia lipolytica*. The level of organization of this work is quite impressive. The manuscript is well written and have the support of a Manual that is an important guide to use and understand the toolkit. In addition, practical applications illustrate the strength of their toolkit by the characterization of 137 promoters and the engineering of homogentisic acid production in *Y. lipolytica*. I believe, the present work is very interesting for those researchers striving to improve their knowledge on DNA recombinant techniques, and certainly will provide a good support for others specifically interested in *Yarrowia*. Therefore, I do support the manuscript publication, and only have minor comments as suggestions for the authors.

Thanks for the positive comments.

In my perception, the abstract is not quite direct about the essence of the work, which is the assembly of a formidable toolkit of 147 plasmids to support the metabolic engineering of *Y. lipolytica*. It is Ok to emphasize aimed improvements, but those come from using the plasmid toolkit, comprising seven specific modules, each one having a functionality. I also think it is important to mention early on that the toolkit is available through the Addgene repository.

Thanks for the suggestion. We have now mentioned early on that the toolkit is available in Addgene: “*The basic set comprises 147 plasmids, two E. coli and three Y. lipolytica strains (Tables S8 and S9 of Supplementary Manual), which are available through Addgene.*”

We have now revised the abstract as suggested by reviewer 2 in order to describe more the modular aspect of the toolkit.

The final part of the new abstract reads as follows:

“As a proof of concept, we developed the YaliCraft toolkit for Yarrowia lipolytica, which is composed of a basic set of 147 plasmids and 7 modules with different purposes. We used the toolkit to generate and characterize a library of 137 promoters and to build a de novo strain synthesizing 373.8 mg/L homogentisic acid.”

I think of some tiny suggestions to improve the understanding of the system described in the Figure 1. A simple label such as “Level 0 (Lvl0): entry vectors” could make clearer that the basal plasmids are the assembly pieces for the upper-level plasmids. When possible, it would be interesting to write a number describing how many plasmids of each kind (pPro, pTer, pInt, etc...) are available as part of the toolkit. I think that the explanations of modules present in the Manual

as captions of Fig S1, could be incorporated in the Figure 1 caption, on the main manuscript. Probably, a reference to the Table S8 in the Figure 1 caption is also important.

Thanks for the suggestions. We have now modified figure 1 and its caption to include the suggested comments: the number of plasmids, the descriptions from Fig S1 and a reference to Table S8.

The new Fig 1 caption reads as follow:

“Figure 1. Modular toolkit structure. The toolkit consists of seven modules for quick and easy assembly of integrative constructs and Cas9-helper plasmids. Lvl0 Module: single parts in entry vectors. Exp Module: assembly of overexpression constructs. Pro Module: assembly and screening of new promoters. Del Module: assembly of disruption constructs. Int Module: changing integration loci by homology arms exchange. MEx Module: assembly of marker-free constructs by selectable marker excision. Cas Module: redirection of Cas9-helper to new genome loci. Five of the modules - Pro, Del, Cas, Int, and MEx Modules - represent a new methodology that functionally extends previously used GG assembly systems. Assembly of Pro, Del and Int Modules are based on single GG reactions, while Cas and MEx Modules involve homologous and site-specific recombination taking place in the special E. coli strains. The arrows between modules indicate different orders in which they can be applied to enable variable genome engineering techniques as shown in the top panel, Yeast. Table S8 is a list of the main plasmids of the toolkit.”

Apart from those minor points, the manuscript seems robust to me, and I believe the readers can learn from it. I congratulate the authors for the good work.

We would like to thank again Reviewer 2 for their time and positive comments and suggestions.

Reviewer #3 (Remarks to the Author):

Technologies based on CRISPR/Cas9 have been developed to facilitate microbial cell engineering, even though they still have some limitations and do not always allow simple and fast genome editing. In their manuscript, the Authors have identified three of those limitations and proposed a corresponding engineering solution. The presented YaliCraft toolkit is based on modules that extend the versatile Golden-Gate assembly system already available in *Y. lipolytica*. Each module comprises a specific type of molecular operation modifying the final vector, namely assembly of new promoters (Pro Module), assembly of gene disruption construct (DelModule), the possibility for quick exchange of homologous arms (HA) used for integration (Int Module), a module allowing the switch from marker free to marker-based integration (MEx Module) and a Cas Module allowing an easy and fast re-encoding of gRNA in Cas9-helper plasmid. As a proof of concept, the Authors have characterized a set of 137 promoters (Pro Module) and engineered *Y. lipolytica* either by gene disruption or overexpression to produced homogentisic acid. Beside the manuscript, the Authors propose a leaflet of 40 pages detailing protocols on how to use the YaliCraft toolkit. The manuscript and the YaliCraft manual are well written and structured. In its concept, this extended toolkit is compatible with those already available for *Y. lipolytica* engineering. This toolkit will be helpful for the *Yarrowia* community. I do not have specific comments on the manuscript.

We appreciate the time taken by reviewer 3 in assessing our manuscript and we thank them for their positive comments about our work.

REVIEWERS' COMMENTS:

Reviewer #1 (Remarks to the Author):

Dear authors,
thank you for the preparation of the well-structured rebuttal letter - it made it very easy to follow your adjustments. I am very happy to see that my comments were helpful. All my comments have been well addressed.

Since the first revision, I noticed two small things in your manual that you might want to quickly address, but that don't require a revision in my opinion. Just fix them before the final submission if possible:

1. The page numbers in the table of contents need to be updated. E.g. "Y. lipolytica colony PCR" is supposed to be on page 34 but is actually on page 37.
2. Line 972+973 of the manual: "this amount of biomass visually correspond to the size of a match head and for the W29 strain it is equal to 1 mL of cell suspension with an OD600 of 20". I like that you put this information but from my experience, a cell pellet of this size should not result in an OD of 20 when resuspended in 1 mL. Could it be that you mean 2.0?

Once again, congratulations on a great project and happy crafting.
OK

Reviewer #2 (Remarks to the Author):

The authors have addressed all my comments and, from my part, the manuscript is ready for publication.

REVIEWERS' COMMENTS:

Reviewer #1 (Remarks to the Author):

Dear authors,
thank you for the preparation of the well-structured rebuttal letter - it made it very easy to follow your adjustments. I am very happy to see that my comments were helpful. All my comments have been well addressed.

Since the first revision, I noticed two small things in your manual that you might want to quickly address, but that don't require a revision in my opinion. Just fix them before the final submission if possible:

1. The page numbers in the table of contents need to be updated. E.g. "Y. lipolytica colony PCR" is supposed to be on page 34 but is actually on page 37.

Thanks for noticing. We have now updated the table of content.

2. Line 972+973 of the manual: "this amount of biomass visually correspond to the size of a match head and for the W29 strain it is equal to 1 mL of cell suspension with an OD600 of 20". I like that you put this information but from my experience, a cell pellet of this size should not result in an OD of 20 when resuspended in 1 mL. Could it be that you mean 2.0?

Thanks for this observation. We have revised this number and in the equipment used this corresponded to an OD600 of 10 and we have modified this accordingly.

Once again, congratulations on a great project and happy crafting.

Thanks for your time and suggestions.

Reviewer #2 (Remarks to the Author):

The authors have addressed all my comments and, from my part, the manuscript is ready for publication.

Thanks for your time and suggestions.